# The Relationship between Training Cycle-Dependent Fluctuations in Resting Blood Lactate Levels and Exercise Performance in College-Aged Rugby Players

**DOI:** 10.3390/jfmk7040093

**Published:** 2022-10-21

**Authors:** Ryotaro Kano, Kohei Sato

**Affiliations:** 1Graduate School of Education, Tokyo Gakugei University, Tokyo 184-8501, Japan; 2Department of Health and Physical Education, Tokyo Gakugei University, Tokyo 184-8501, Japan

**Keywords:** rugby players, de-training, training periodization, athlete

## Abstract

An increase in resting blood lactate (La^−^) concentration due to metabolic conditions has been reported. However, it is not clear whether resting La^−^ changes with training cycles in athletes. The purpose of this study was to test the hypotheses that (1) the morning resting La^−^ levels are lower in periods of high training compared to periods of low training and (2) these changes in La^−^ concentration are related to athletes’ metabolic capacity during exercise in male college-aged rugby players. Resting La^−^ and blood glucose concentrations were measured in the morning in eight league rugby players during the summer pre-season period (Pre-period), the training and competition season period (TC-period), and the winter post-season period (Post-period). In each period, anaerobic power, La^−^ concentration, and respiratory responses were measured during the 40 s maximal Wingate anaerobic test (WT). The resting La^−^ concentration in the morning was significantly lower in the TC-Period (1.9 ± 0.6 mmol/L) than in the Post-Period (2.3 ± 0.9 mmol/L). The rate of decrease in La^−^ level immediately after the 40 s WT was significantly higher in the TC-Period than in the Post-Period. The resting La^−^ concentration was significantly correlated with the peak oxygen uptake and the carbon dioxide output during the WT. These results support the hypothesis that an athlete’s training cycle (i.e., in season and off season) influences the resting La^−^ levels as well as the metabolic capacity during high-intensity exercise. The monitoring of resting La^−^ fluctuations may provide a convenient indication of the training cycle-dependent metabolic capacity in athletes.

## 1. Introduction

Resting blood lactate concentrations (La^−^) may be systematically impacted by metabolic diseases. Specifically, the resting La^−^ levels tend to be higher in people with diabetes, obesity, and chronic fatigue [1]. It is also notable that mitochondrial mass and function are suppressed by diabetes mellitus, which reduces the capacity for oxidative ATP production [2,3]. Consequently, the dependence on glycolytic ATP production as a compensatory effect is accompanied by an increase in resting La^−^ levels [4]. Therefore, the resting La^−^ concentration might be utilized as a diagnostic indicator of metabolic diseases. In addition, athletes’ metabolic conditioning varies greatly with the amount and intensity of exercise during different competitive seasons. The resting La^−^ level may be used as an indicator of an athlete’s physical readiness throughout athletes’ competitive seasons.

The resting La^−^ concentration in athletes has not received much attention as a metabolic biomarker. The range of alterations in resting La^−^ levels is considerably smaller than that during exercise. Athletes routinely use La^−^ concentration during exercise as a measurement of exercise intensity level [5,6]. In athletes, mitochondrial structure and function respond rapidly across training cycles, which in turn determines their metabolic capacity [7,8,9]. The amount of lactate membrane transport proteins (MCT1 and MCT4) and glucose transporters (GLUT4) also vary with the training conditions [10]. In particular, exercise training types such as high-intensity interval training (HIIT) have been shown to enhance the mitochondrial function [11]. Lactate has also been reported to be a signaling molecule that enhances mitochondrial biogenesis [12,13]. Lactate circulating in the blood plasma is effectively used as an energy substrate for various organs [7]. In athletes with increased skeletal muscle mitochondrial capacity, the lactate-utilizing capacity of muscle fibers is increased, which may affect both exercising and resting lactate dynamics. In fact, a one-year case study from our laboratory reported that resting La^−^ in one track sprinter fluctuated significantly in response to the imposition of training and conditioning cycles [14]. Compared to the de-training phase (prolonged rest due to injury), the training phase showed significantly lower resting La^−^ when measured in the morning. However, the relationship between training cycles and resting La^−^ concentration, to our knowledge, has not been explored in other situations in relation to athletes.

To address this issue, rugby players were selected, in part, because rugby has a well-defined training cycle. As indicated by the World Rugby Union’s training guidelines, rugby player training is characterized by periodization (Pre-season, In-season, and Off-season). In the Pre-season, the players are required to improve the fundamentals of strength and endurance. In the in-season period, the players maintain their physical fitness level and prepare for competitions by practicing technical and tactical skills. The off-season allows for an adequate recovery from chronic fatigue and the preparation for the next season.

Therefore, the purpose of this study was to test the hypothesis that (1) morning resting La^−^ levels are lower in periods of high training compared to periods of low training and (2) these changes in La^−^ concentration are related to the metabolic capacity during exercise in male college-aged rugby players. The findings of this study could be applied to athletes in the future, as blood lactate concentration at rest could be used as a biomarker of metabolic conditioning.

## 2. Materials and Methods

### 2.1. Subjects

A total of 8 well-trained Tokyo Rugby Union League players (age: 20.6 ± 2.1 years, height: 172.8 ± 9.4 cm, rugby position: 3 forwards and 5 backs) participated in this study. All protocols were explained, and informed consent was obtained from all individual participants. The study was approved by the Tokyo Gakugei University Ethics Board (No. 506). The subjects completed a health questionnaire and declared that they had no history of metabolic diseases, musculoskeletal injuries, medications, or smoking habits in the past year.

### 2.2. Experimental Design

An overview of the experimental design of this study is presented in Figure 1. The training period cycle was divided into (1) a summer pre-season period (Pre-period: 2021, July), (2) a training and competition-season period (TC-period: 2021, August–December), (3) a winter post-season period (Post-period: 2022, January–March).

Training content in the Pre- and Post-periods: individual resistance training was the main focus, and field training (running, passing, mini-games, etc.) was not conducted. Resistance training, intensity: 80% of 1RM for 8 to 10 repetitions, number of sets: 4 sets, type of lift: bench press, squat, arm curl, bent-over row, deadlift, leg curl, side raise, and chinning. Frequency: Pre-period, 3–6 times/week; Post-period, 0–3 times/week.

Training content in the TC-period: team practice was conducted for 5 days/week, approximately 2 h/day.

Warm-up (jogging, sprinting drills) for 15 min.

Fitness training (circuit exercises, high-intensity short interval running) for 5 min.

Contact training (tackle-based exercise, breakdown formations) for 15 min.

Handling training (passing and kicking skills practice) for 15 min.

Game match style formations (forwards and backs combinations) for 30 min.

Weight training was conducted as individual practice 3–6 times/week.

In the last week of each period, immediately after waking in the morning, blood samples were taken by puncturing the tip of the left middle finger (Naturalet EZ, Arkray, Kyoto, Japan). Blood lactate and glucose levels were determined by a lactate analyzer (Lactate Pro2, Arkray, Japan) and a glucose analyzer (PG-7320, Arkray, Japan), respectively. No hemolytic agents were used. We used 0.3 µL and 0.6 µL of blood from the fingertip of the hand for the Lactate Pro2 and PG-7320 analyzers, respectively. Following this sampling period, an exercise performance test was conducted to evaluate the physiological characteristics of each cycle (Figure 1).

### 2.3. Exercise Performance Measurement

All exercise performance measurements during each period were conducted in the same laboratory environment (room temperature 23–25 °C). The subjects avoided any strenuous exercise 24 h prior to the test. The participants consumed a meal at least 3 h before the test and avoided caffeine intake.

After body composition was measured, La^−^ concentration, heart rate, and respiratory parameters were measured in a seated resting position. Exercise performance capacity and physiological characteristics were assessed by the 40 s Wingate anaerobic test (7.5% body weight load, 40-s WT) using a bicycle ergometer (Power-max V3, Konami Sports, Tokyo, Japan). After warming up for 3 min, the 40-s WT was performed. After the 40-s WT, pedaling was maintained continuously for 3 min. Subsequently, La^−^ concentration (after 3, 10, and 20 min from the test) and respiratory measurements were continuously made in the supine position.

### 2.4. Body Composition

Body weight, body fat percentage, and skeletal muscle mass were measured using a dual-type body composition analyzer (Inner Scan Dual RD-801, Tanita Corporation, Tokyo, Japan). The participants were instructed to only wear their undergarments to minimize the measurement error.

### 2.5. Respiratory Measurements

Respiratory gas analysis was performed using an open-circuit automatic O_2_ and CO_2_ analyzer with a hot-wire flowmeter (AE300S, Minato Medical Science, Tokyo, Japan). Before each measurement, the gas analyzer was calibrated twice using a commercial gas (Sumitomo Seika, Tokyo, Japan) for the precision measurement of O_2_ and CO_2_ concentrations.

### 2.6. Calorie Calculation

The subjects were asked to report their weekly dietary intake at the time of the blood sampling. Caloric intake was calculated using the calorie table provided by the Ministry of Agriculture, Forestry and Fisheries of Japan.

### 2.7. Statistical Analysis

All statistical data are expressed as mean ± standard deviation (SD). Two-way analysis of variance was used for La^−^ and glucose levels at waking in the morning on each training cycle (training cycle × interindividual variation) and for changes in La^−^ levels and respiratory gas during the 40-s WT (training cycle × time series). Changes in body composition, mean power, maximal power, and La^−^ concentration in the 40-s WT were examined by one-way analysis of variance, and the Tukey–Kramer’s multiple comparison test was used as a post hoc test. The Pearson’s correlation coefficient was employed for the correlation analysis. Statistical significance was set at *p* < 0.05. All statistical analyses were performed using Prism software (Version 9, GraphPad Software, San Diego, USA).

## 3. Results

### 3.1. Whole Body Composition

The body composition and daily caloric intake in each period are shown in Table 1. There was a significant increase in skeletal muscle mass (kg) and a significant decrease in body fat percentage (%) in the TC-Period compared to the Post-Period. The body weight and average caloric intake did not change throughout each period.

### 3.2. Resting Blood Lactate and Glucose Levels in the Morning (after Waking Up)

The changes in resting La^−^ and blood glucose levels measured continuously for one week in each period are shown in Figure 2. The resting La^−^ level during the TC-period (1.9 ± 0.6 mmol/L) was lower than that during the Post-Period (2.3 ± 0.9 mmol/L, *p* = 0.024) and tended to be lower than during the Pre-period (2.2 ± 0.5, *p* = 0.079). The resting blood glucose levels did not change among periods (Pre-period: 96.4 ± 6.1 mg/dL, TC-period: 94.3 ± 4.8 mg/dL, Post-period: 96.0 ± 1.7 mg/dL).

### 3.3. Analysis of the 40 s Wingate Anaerobic Test (40-s WT) 

The results of the 40-s WT (average and peak power) are shown in Figure 3. There were no significant differences in the 40-s WT results between the Pre-Period and the TC-Period. The mean power values in the Post-Period were lower than in the Pre-Period (−5.3%, *p* = 0.006) and tended to be lower than in the TC-Period (−3.8%, *p* = 0.063). The peak power tended to be higher in the TC-Period than in the Post-Period (*p* = 0.056), but there was no significant difference among the three periods.

Figure 4 shows La^−^ levels at rest and 3, 10, and 20 min after the 40-s WT. There were no significant differences in La^−^ concentration during the resting and recovery phases among the three periods. However, the rate of La^−^ decrease from 10 to 20 min was significantly faster in the TC-Period (34.6 ± 5.1%) than in the Post-Period (20.8 ± 11.8%).

Respiratory gas parameters: Figure 5 shows peak oxygen uptake and carbon dioxide output before the 40-s WT (5 min average) and during pedaling (every 10 s) and the recovery phase (every 30 s). The two-way analysis of variance showed no significant interaction between the periods.

Relationship between resting La^−^ concentration in the morning and 40-s WT parameters: the mean resting La^−^ level during one week in each period showed a correlation with peak VO_2_ (R = −0.60, *p* = 0.002) and peak VCO_2_ (R = −0.62, *p* = 0.001) during the 40-s WT (Figure 6). Small non-significant correlations were found between mean resting La^−^ levels and mean power during the 40-s WT (R = −0.38, *p* = 0.067).

## 4. Discussion

The present study revealed that fluctuations in resting La^−^ levels at the time of waking might constitute a biomarker for estimating the levels of and changes in metabolic capacity in rugby players across their on- and off-seasons. The main results are as follows: (1) The resting La^−^ concentration was significantly higher during the Post-period (rest and conditioning term) than during the TC-period (high-intensity exercise training period); (2) this La^−^ response demonstrated a significant correlation with peak oxygen uptake and carbon dioxide output during the 40-s WT. The physiological implication of these results is that changes in training cycle-dependent metabolic capacity are reflected in resting La^−^ levels.

It is well known that physical training increases the lactate oxidative capacity during exercise as well as exercise performance [8,15,16,17,18,19]. In contrast, patients with metabolic syndrome have reduced lactate oxidative capacity during exercise [20]. The lactate oxidative capacity is mainly determined by the mitochondrial utilization of energy substrates and lactate transport-related membrane proteins. Adenosine triphosphate (ATP) production capacity and oxidative enzyme activity, as mitochondrial functional potentials, increase with training and decrease with de-training [21]. For example, it has been reported that mitochondrial ATP-producing capacity decreases by 12–28% during a 3-week de-training period following 6 weeks of endurance training [22]. A series of studies by Coyle et al. [23,24] demonstrated that citrate synthase activity in endurance-trained subjects decreased from 10.0 to 7.7 (mol/kg protein/h) during the first 3 weeks of de-training. In the training cycles of the subjects in this study, endurance training was not included in the Pre-period and Post-Period, which were 3 and 6 weeks long, respectively. Considering the changes in mitochondrial function due to detraining, mitochondrial function during the Pre- and Post-Periods was estimated to be considerably reduced. However, there was no significant difference in the mean and peak power of the 40-s WT between the Pre- and the TC-periods (Figure 3). One of the reasons for this result may be related to the implementation of resistance training during the Pre-period. It has been reported that resistance training (3 times per week for 12 weeks) increases the levels of mitochondrial proteins and some transcripts in young healthy male subjects [25]. In the Pre-period, resistance training at least 3 times per week may have maintained the WT performance despite the lack of endurance training. In contrast, the Post-period, in which resistance training was performed less frequently (0–3 times per week), tended to decrease the WT performance compared to the TC-period (Figure 3).

The monocarboxylate transporters (MCTs) in skeletal muscle mainly include MCT1, which transports blood lactate into the muscle fibers, and MCT4, which releases lactate from the muscle fibers into the blood. It has been suggested that the amount of MCT1 correlates with the removal rate of blood lactate after intense exercise [15]. The levels of MCT1 and MCT4 are increased by various types of training (endurance training, sprint high-intensity interval training, resistance training) [16] or decreased by de-training. Following sprint interval training, the MCT 1 and MCT4 protein levels are significantly reduced during a one-week de-training period [10]. One previous study in elite athletes suggested that high-intensity training is necessary to maintain the MCT levels [26]. Considering these previous studies, it is possible that the metabolic capacity for lactate during exercise was reduced in the Post-period compared to the TC-period. In fact, the removal rate of blood lactate (from 10 to 20 min) after the 40-s WT showed a significant attenuation in the Post-period compared to the TC-period (Figure 4).

Increased fasting plasma lactate concentrations have been identified as a clinical finding in patients with cardiometabolic diseases [27,28,29]. The mechanism by which fasting plasma lactate concentrations increase in patients with cardiometabolic diseases is thought to be a compensatory increase in glycolysis due to the reduced aerobic substrate oxidation capacity associated with decreased mitochondrial function and quantity [1]. Recently, a negative correlation between fasting La^−^ levels and lipid oxidation capacity and endurance exercise capacity was reported in a study on healthy women without metabolic disease [30]. This study points to the usefulness of the resting La^−^ concentration as a biomarker for the early detection of metabolic diseases. One case study, from our laboratory, focusing on the resting blood lactate variability in sprint athletes reported a coupling between a one-year training cycle and resting (waking) blood lactate levels [14]. A common finding in these studies of patients with metabolic diseases, adult women, and sprinters is that the resting (fasting or waking) blood lactate levels are elevated due to a decreased metabolic capacity. In support of these findings, the resting (waking up) La^−^ concentration was higher in the Post-period when mitochondrial function was expected to be simply downregulated. In addition to the authors’ case study [14], a training cycle dependence of the resting (immediately after waking up in the morning) La^−^ levels was found in rugby players herein. Furthermore, in the present study, there was a significant correlation between peak VO_2_ and VCO_2_ in the 40-s WT and the resting La^−^ level at waking. The dynamics of VO_2_ and VCO_2_ during maximal exercise are performance-determining factors [31,32], suggesting mitochondrial functional properties.

In summary, the resting La^−^ levels in college rugby players at the time of waking were higher in the off-season than in the training season, suggesting that the resting lactate concentration may be a potential biomarker reflecting the metabolic capacity. A limitation of this study is that the mitochondrial and MCTs protein levels were not directly measured due to the difficulty in obtaining muscle tissue samples from active athletes. Future investigations might usefully address these knowledge gaps in order to clarify the mechanisms by which the resting La^−^ levels fluctuate in a training cycle-dependent manner. Furthermore, it is necessary to examine whether the same phenomenon occurs in endurance athletes, such as marathon runners, in whom the mitochondrial function is of overarching importance.

## Figures and Tables

**Figure 1 jfmk-07-00093-f001:**
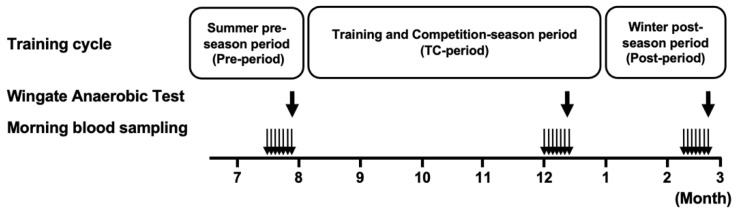
Schematic overview of the experimental design. The bold arrows indicate the time of the 40 s Wingate test and metabolic measurement during each period. The narrow arrows indicate blood lactate sampling at the time of waking each day of the week.

**Figure 2 jfmk-07-00093-f002:**
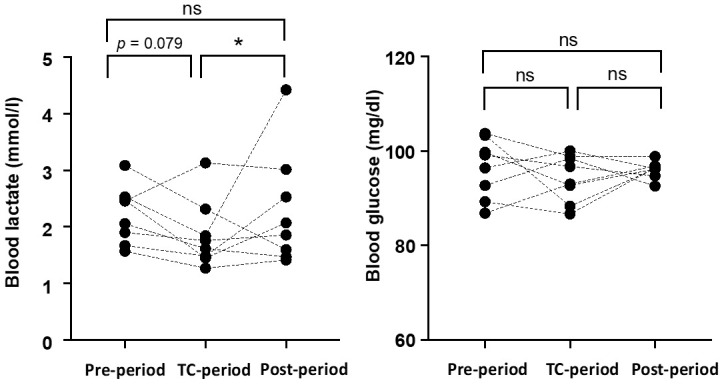
Changes in morning blood lactate and blood glucose levels in each period. The dots represent the average values for the subjects during one week. * *p* < 0.05 vs. the Post-period.

**Figure 3 jfmk-07-00093-f003:**
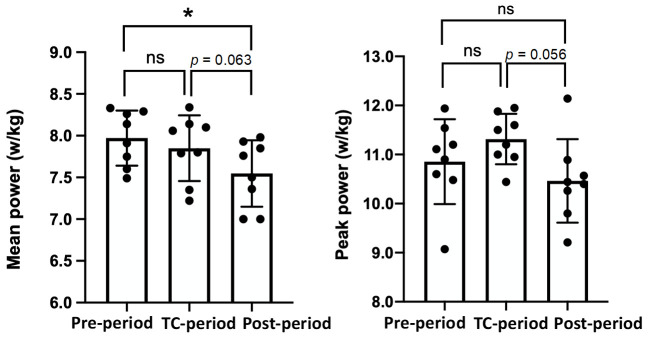
Mean and peak cycling power (w/kg) during a 40 s Wingate test in each period. The values are expressed as means ± SD. * *p* < 0.05 vs. the Post-period.

**Figure 4 jfmk-07-00093-f004:**
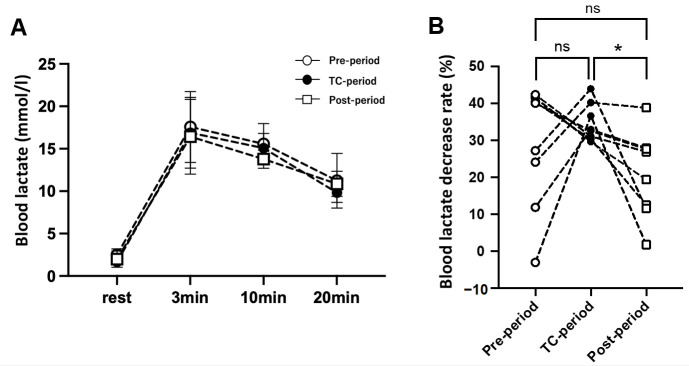
(**A**), Blood lactate levels (mmol/L) before (rest) and after (3, 10, and 20 min) the 40 s Wingate test in each period. (**B**), Percentage decrease in blood lactate level during the recovery phase from 10 to 20 min. The values are expressed as means ± SD. * *p* < 0.05 vs. the Post-period.

**Figure 5 jfmk-07-00093-f005:**
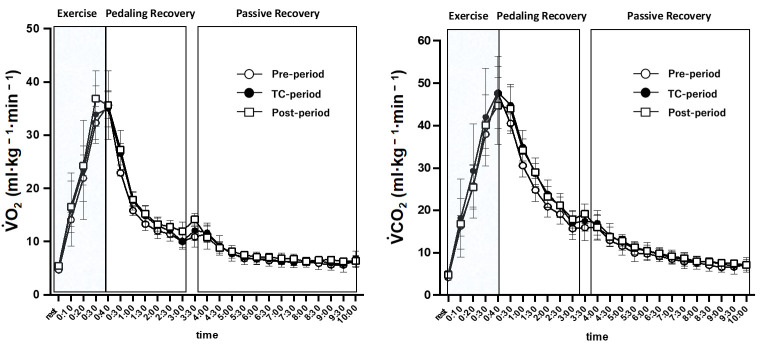
VO_2_ (ml·kg^−1^·min^−1^) and VCO_2_ (ml·kg^−1^·min^−1^) during and after the 40 s Wingate test in each period. The values are expressed as means ± SD.

**Figure 6 jfmk-07-00093-f006:**
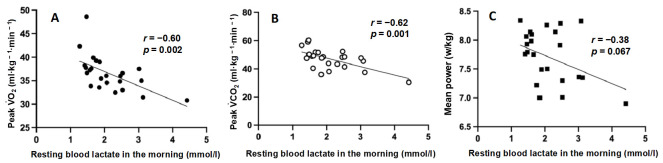
Relationship between resting blood lactate in the morning and peak VO_2_ (**A**), peak VCO_2_ (**B**), mean power (**C**) during 40 s Wingate test. The dots represent the average values for the subjects during one week in each period.

**Table 1 jfmk-07-00093-t001:** Comparison of whole body composition and caloric intake in each period. The values are expressed as means ± SD. * *p* < 0.05 vs. the Post-period values.

	Pre-Period	TC-Period	Post-Period
Body Weight (kg)	77.8 ± 8.9	78.6 ± 10.0	77.8 ± 10.9
Muscle Mass (kg)	34.4 ± 4.2	35.2 ± 4.4 *	34.2 ± 4.3
Body Fat (%)	22.3 ± 3.6	21.5 ± 4.1 *	22.7 ± 4.4
Calorie Intake (kcal/day)	2426.3 ± 443.8	2543.1 ± 431.4	2657.1 ± 454.1

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
