# Peer review of "The Relationship between Training Cycle-Dependent Fluctuations in Resting Blood Lactate Levels and Exercise Performance in College-Aged Rugby Players"

_jfmk, 2022, doi:10.3390/jfmk7040093_

Round 1

Reviewer 1 Report

General comments

This manuscript aims at testing the hypotheses that 1) morning resting blood lactate concentrations [La-] are lower in periods of high training compared to periods of low training and 2) these changes in [La-] are related to metabolic capacity during exercise in male college-aged rugby players. Despite some minor issues detailed below, authors manage sufficiently to fulfill their aims.

Minor comments

(line 13) … ]) are…

(l149) please, do not use acronyms in headings;

(l185) … changes in metabolic…

(l196 and elsewhere throughout MS) please, do not start sentences with acronyms;

(l207 and 214) … Figure 3…

(l211-2) please, re-phrase;

(l227) … Figure 4…

(l257÷60 and 262-3) ??

Author Response

Thank you for your comments on this study and for pointing out areas for revision. I have corrected all the points you pointed out, including careless mistakes, style, etc.

(line 13) … ]) are…

Thank you for pointing this out. We have corrected it. (Line 13)

(l149) please, do not use acronyms in headings;

Thank you for pointing this out. We have corrected it. (Line 164)

(l185) … changes in metabolic…

Thank you for pointing this out. We have corrected it. (Line 219)

(l196 and elsewhere throughout MS) please, do not start sentences with acronyms;

Thank you for pointing this out. We have corrected it. (Line 230)

(l207 and 214) … Figure 3…

Thank you for pointing this out. We have corrected it. (Line 241 ; 248)

(l211-2) please, re-phrase; 

Thank you for pointing this out. We have modified the sentences as follows: “In the Pre-period, resistance training at least 3 times per week may have maintained WT performance despite the lack of endurance training. “(Line 245-246)

(l227) … Figure 4…

Thank you for pointing this out. We have corrected it. (Line 261)

(l257-60 and 262-3) ??

The submission format remained and was deleted. I apologize for the careless mistake.

Reviewer 2 Report

General: change “[La-]” to “La-“. Brackets make it hard to read.   

What is the term “training status”? Please define training status.  Training status is a broad term like “acute, chronic training response or current physical readiness etc”.

Abstract

Page 1

line 13: Space will be added before “are”

Introduction

line 29 to 35: It may need to add more information specifically about why La is a potential indicator for training status. In this paragraph, the authors only explained how La relates to diabetes and ATP production. If you add sentences about how increased La interferes with glycolytic ATP production, associating with sports performance or training status.  

Line 37-44 and page 2 line 50: These sentences may be added into the first paragraph, which may help to connect the sentences in the first paragraph.  The authors explained well why La- is associated with performance.

Page 2

I recommend creating a paragraph what the training load looks like in college-aged rugby players, otherwise, readers can get an idea of how metabolic and neuromuscular fatigue and recovery look like in (college-aged) rugby players.

Line 53-59: I recommend creating a new paragraph 1) information is lacking as you said in lines 53-55, 2) how the information will help sports science /strength and conditioning community, and then  3) purpose statement and hypothesis. Also, where is your purpose statement? I don’t know what the authors tried to do in this study.

Line 63-67: any inclusion or exclusion criteria?

Line 75-77: Maybe helpful to use a table to summarize how the training program was implemented. Also, what was the intensity each week?  The sentence made me think authors provided 4 sets of 8-10 RM, which doesn’t seem right. For example, 8 RM is the maximum amount of weight you can lift for a single set. So, please provide precise terms. Additionally, what were the volume loads during pre-, TC- and post-period?

Line 78 -79: do not create a paragraph of a single sentence. Connect with the previous paragraph. Also, what is the weekly training load during TC-period?

Line 80: Authors cannot leave a noun without a verb. Reworded.

Line 85-89: Should include average and last 7 days average of training loads. The volume and intensity will affect La and lab performance measurements.

Page 3

line 96-116: Please list performance, body composition, reparatory and calorie measurements in order. This made me think that authors performed Wingate test first, followed by body composition, and other measurements.

Line 118-119: this should be one-way repeated ANOVA because the authors examine the difference of each time point. What factors do you put for two-way ANOVA (period (or training cycle) and ???)

Page 4

Line 141: reword this sentence.

Line 144: spell out the exact p-value instead of “(p<0.05).

Line 150: remove “Exercise performance:”

Line 152: same as line 144.

Line 156:  remove “Blood lactate:”

Page 5

Line 176: you spelled out the exact p-value when significant. Keep it consistent.

Page 6

Discussion:

The biggest issue of this study was what was training load during each period.  The hypothesis is morning resting LA is lower than in other periods, but how do reviewers and readers know these results are repeatable without the information? For example, if subjects completed a hard gym session and game before measurement days, it will affect La and performance results? How can you control that and what was the value of training loads?

Line 186-188:  overstatement since authors did not show any objective values in internal and external training loads.

Line 193-194: this study did not deal with a metabolic syndrome so remove it.

Line 204: this study did not examine mitochondrial function, so this is an overstatement.

Line 207-210:  When stating that, provide how authors periodize training programs. (e.g., week 1 at 80% of 8RM, week 2 at 85% of 8RM etc.) If you don’t provide any information, this study is not repeatable.

Line 215-223: None of them are examined in this study, so I am not convinced that it happened in this study.

Page  7

Line 257-263. Remove lines 257 to 263.

Author Response

Comments and Suggestions for Authors

General: change “[La-]” to “La-“. Brackets make it hard to read.  

→Thanks for pointing this out. All the words have been corrected.

What is the term “training status”? Please define training status.  Training status is a broad term like “acute, chronic training response or current physical readiness etc”.

→ As you pointed out, status has a broad meaning. We have replaced it with “physical readiness”. (Line 37)

Abstract

Page 1

line 13: Space will be added before “are”

→Thank you for pointing this out. We have corrected it. (Line 13)

Introduction

line 29 to 35: It may need to add more information specifically about why La is a potential indicator for training status. In this paragraph, the authors only explained how La relates to diabetes and ATP production. If you add sentences about how increased La interferes with glycolytic ATP production, associating with sports performance or training status.  

Line 37-44 and page 2 line 50: These sentences may be added into the first paragraph, which may help to connect the sentences in the first paragraph.  The authors explained well why La- is associated with performance.

→ In consideration of these comments and suggestions, we have reviewed the description of each paragraph.

The following comments have been included in the first paragraph to make it easier to understand the connection between the content of the first and second paragraphs. (Line 34-38)

“Therefore, resting La- might be utilized as a diagnostic indicator of metabolic diseases. In addition, athletes' metabolic conditioning varies greatly with the amount and intensity of exercise during different competitive seasons. Resting La- may be used as an indicator of an athlete's physical readiness throughout athletes’ competitive seasons.”

Page 2

I recommend creating a paragraph what the training load looks like in college-aged rugby players, otherwise, readers can get an idea of how metabolic and neuromuscular fatigue and recovery look like in (college-aged) rugby players.

 →Thank you for your valuable comments. Therefore, we have included the training characteristics and cycle dependence of rugby players in the introduction as follows : ( line 55-63 )

“As indicated by the World Rugby Union's training guidelines, Rugby player training is characterized by periodization (Pre-season, In-season and Off-season). In the Pre-season, the player is required to improve the fundamentals of strength and endurance. In-season, the players maintain their physical fitness level and prepare for competitions by practicing technical and tactical skills. The off-season is for adequate recovery from chronic fatigue and preparation for the next season.”

Line 53-59: I recommend creating a new paragraph 1) information is lacking as you said in lines 53-55, 2) how the information will help sports science /strength and conditioning community, and then  3) purpose statement and hypothesis. Also, where is your purpose statement? I don’t know what the authors tried to do in this study.

→Thank you for your helpful comments. The following new paragraphs have been set. (Line 64-68)

“Therefore the purpose of this study was to test the hypothesis that 1) morning resting La- are lower in periods of high training compared to periods of low training and 2) these changes in La- are related to metabolic capacity during exercise in male college-aged rugby players. This finding could be applied to athletes in the future, as blood lactate concentration at rest can be used as a biomarker for metabolic conditioning.”

Line 63-67: any inclusion or exclusion criteria?

→If any one of these criteria was met within the past year, the subject was excluded. These comments were added to the text. (Line 76-77)

Line 75-77: Maybe helpful to use a table to summarize how the training program was implemented. Also, what was the intensity each week?  The sentence made me think authors provided 4 sets of 8-10 RM, which doesn’t seem right. For example, 8 RM is the maximum amount of weight you can lift for a single set. So, please provide precise terms. Additionally, what were the volume loads during pre-, TC- and post-period?

 →The detailed description of training has been modified. The intensity of resistance exercise is 80% of 1RM for 4 sets of 8 to 10 repetitions. We have corrected it. (Line 85)

The training volume during each cycle can be roughly estimated from the duration and intensity of the exercise, but here we have provided specific training details. The training content was presented in text, although we also considered presenting it as a new table. Thanks for your advice.

Line 78 -79: do not create a paragraph of a single sentence. Connect with the previous paragraph. Also, what is the weekly training load during TC-period?

→In this study, we are not able to show specific values for training load during team practice because we were not able to measure heart rate or mileage. However, as you pointed out, we thought it would be better to show the content of the training, We have summarized examples of typical TC period training in Methods. (Line 88-95)

Line 80: Authors cannot leave a noun without a verb. Reworded.

→The relevant paragraphs were changed. (Line 90-95)

Line 85-89: Should include average and last 7 days average of training loads. The volume and intensity will affect La and lab performance measurements.

→As you indicated, the lab performance test may be affected by training activity of the previous few days. However, the average volume and intensity for each cycle period and the last 7 days were planned to remain constant. Therefore, the effect of the sampling period on the La - and lab performance measurements is expected to be small. In addition, 24 hours prior to the lab performance measurement, the subjects were instructed not to perform strenuous exercise. Thus, any temporary influence on lab performance was eliminated.

Page 3

line 96-116: Please list performance, body composition, reparatory and calorie measurements in order. This made me think that authors performed Wingate test first, followed by body composition, and other measurements.

→Thank you for pointing this out. We have corrected it. (Line 110-116)

Line 118-119: this should be one-way repeated ANOVA because the authors examine the difference of each time point. What factors do you put for two-way ANOVA (period (or training cycle) and ???)

→Thank you for pointing this out. To include the individual variability in each cycle in the statistics, we adapted a statistical analysis with Two-way ANOVA.

Page 4

Line 141: reword this sentence.

→Thank you for pointing these out. We have corrected it. (Line 155-158)

Line 144: spell out the exact p-value instead of “(p<0.05).

→Thank you for pointing these out. We have corrected it. (Line 157)

Line 150: remove “Exercise performance:”

→Thank you for pointing these out. We have removed it.

Line 152: same as line 144.

→Thank you for pointing these out. We have corrected it. (Line 158)

Line 156:  remove “Blood lactate:”

→Thank you for pointing these out. We have removed it.

Page 5

Line 176: you spelled out the exact p-value when significant. Keep it consistent.

→Thank you for pointing these out. We have corrected it. (Line 210)

Page 6

Discussion:

The biggest issue of this study was what was training load during each period.  The hypothesis is morning resting LA is lower than in other periods, but how do reviewers and readers know these results are repeatable without the information? For example, if subjects completed a hard gym session and game before measurement days, it will affect La and performance results? How can you control that and what was the value of training loads?

→Quantifying the load of each training cycle is important to confirm the validity and reproducibility of the findings of this study. In a contact sport such as rugby, it is difficult to quantify physical activity by evaluating parameters such as heart rate variability and running distance. However, the total training load in rugby can be clearly differentiated for each cycle. Therefore, this fact would lead us to conclude that the findings of this study are dependent on the training cycle. A direct causal relationship between the results of this study (waking resting lactate, exercise performance) and training activity (e.g., muscle strength, sprinting, endurance, etc.) cannot be determined.

Line 186-188:  overstatement since authors did not show any objective values in internal and external training loads.

→As mentioned above, we have not been able to demonstrate training load values, but the low resting blood lactate concentrations at morning were limited to the TC-period. We believe it is reasonable to present these findings.

Line 193-194: this study did not deal with a metabolic syndrome so remove it.

→Certainly this study does not address metabolic syndrome. However, previous studies have suggested that glucose and lipid metabolic oxidative capacity is reduced in patients with metabolic syndrome and that resting blood lactate concentration is increased due to dependence on the glycolytic system. Considering these findings, we hypothesized that differences in metabolic capacity during the training-period and off-period would influence resting blood lactate concentrations.

Line 204: this study did not examine mitochondrial function, so this is an overstatement.

→This study did not directly measure mitochondrial morphology or functions. In this paragraph, we referred to the implication of mitochondria as a factor in the decline of metabolic function. The description of the involvement of mitochondrial function is minimal. We did not delete this sentence.

Line 207-210:  When stating that, provide how authors periodize training programs. (e.g., week 1 at 80% of 8RM, week 2 at 85% of 8RM etc.) If you don’t provide any information, this study is not repeatable.

→Thank you. We speculate that the frequency of resistance training during each training period was involved (Pre-period, 3-6 times per week vs Post-period, 0-3 times per week). With the exception of resistance training frequency, other factors were similar between the two periods.

Line 215-223: None of them are examined in this study, so I am not convinced that it happened in this study.

→Certainly the protein content of MCTs was not measured in this study. However, related previous studies have shown that the amount of MCTs decreases with decreasing training volume and that the rate of decrease in blood lactate concentration after high-intensity exercise is correlated with MCT1. Therefore, we did not delete these statements.

Page 7

Line 257-263. Remove lines 257 to 263.

→The submission format remained and was deleted. I apologize for the careless mistake.

Reviewer 3 Report

First line of the abstract seems to suggest that there is mechanistic similarity between lactate observations in disease and physical training. Please reconsider your first line of the abstract.

L23. Please reconsider whether detraining is the best phrase as it seems to the participants where exercising in all cycles.

L60. Delete “old”.

L88. Is there use of lysing agent by the measurement of lactate by the lactate pro 2. Please provide a bit more detail to be sure that blood lactate and not plasma lactate was measured.

L88. Please clarify that the exercise was not immediately after sampling but 3 hr later.

L89. The authors need to provide evidence that an exercise test can provide physiological characteristics of a cycle that can last months.

Ls 117-122. Please clarify whether there was analysis of absolute value as it seems to indicate only changes were analysed. In addition, was there averaging of the lactate samples? Please clarify.

L145. Please be consistent with decimal places for glucose mean and SD values.

In the results section, statements based on p>0.05 are inconsistent with some interpreting them as a difference and some as a trend for a difference. Please revise.

L171. Units for VO2 and VCO2 is ml / (kg min) or ml·kg-1·min-1

In figure 6, p in small and capital font. Please change.

L245 and throughout the manuscript. Please ensure the dot is in the V as it seems to be next to it.

Ls 257-263. Please delete.

Author Response

Comments and Suggestions for Authors

First line of the abstract seems to suggest that there is mechanistic similarity between lactate observations in disease and physical training. Please reconsider your first line of the abstract.

→As you pointed out, we have corrected the sentence to avoid misunderstanding. We have replaced it with the following sentences. “An increase in resting blood lactate concentration ([La-]) due to metabolic condition has been reported.” (Line 10-11)

L23. Please reconsider whether detraining is the best phrase as it seems to the participants where exercising in all cycles.

→We considered “conditioning” to be more appropriate than “detraining” in the meaning of this sentence. (Line 23)

L60. Delete “old”.

→We have corrected it.

L88. Is there use of lysing agent by the measurement of lactate by the lactate pro2. Please provide a bit more detail to be sure that blood lactate and not plasma lactate was measured.

→Thank you for pointing this out. No hemolytic agents are used. Blood from the fingertips is used immediately for the measurement, so hemolysis is not considered a concern; the Lactate Pro uses 0.3 µL of blood from the fingertips of the hand; the PG-7320 uses 0.6 µL of blood from the fingertips of the hand; and the PG-7320 uses 0.5 µL of blood from the fingertips of the hand. These explanations have been added to the text. ( Line 99-102 )

L88. Please clarify that the exercise was not immediately after sampling but 3 hr later.

→As presented in Figure 1, the exercise performance tests were performed on different days from the morning blood sampling.

L89. The authors need to provide evidence that an exercise test can provide physiological characteristics of a cycle that can last months.

→Thank you for pointing this out. Previous studies that divided a one-year training cycle into four periods have shown the validity of measuring exercise performance at the end of each period (MichaÅ‚ et al. 2021). These measurement schedules have been used in similar studies of training periodization.

Ls 117-122. Please clarify whether there was analysis of absolute value as it seems to indicate only changes were analysed. In addition, was there averaging of the lactate samples? Please clarify.

L145. Please be consistent with decimal places for glucose mean and SD values.

→Thank you for pointing this out. We have corrected this issue. (Line 159-160)

In the results section, statements based on p>0.05 are inconsistent with some interpreting them as a difference and some as a trend for a difference. Please revise.

→Thank you for pointing this out. We have corrected this issue. (Line 157-158, 168)

L171. Units for VO2 and VCO2 is ml / (kg min) or ml·kg-1·min-1

→Thank you for pointing this out. We have corrected this issue. (Line 206 and Figure5)

In figure 6, p in small and capital font. Please change.

→Thank you for pointing this out. We have corrected this issue. (Figure 6)

L245 and throughout the manuscript. Please ensure the dot is in the V as it seems to be next to it.

→Thank you for pointing this out. We have corrected these issues. (Line 206, 209, 210, 214, 279, 280)

Ls 257-263. Please delete.

→The submission format remained and was deleted. I apologize for the careless mistake.

Round 2

Reviewer 2 Report

No further edits are needed. Thank the authors for addressing as many edits as they can. 

Author Response

We are very grateful to you for your careful peer review of our paper. In the future we would like to be able to present our data more quantitatively.

Reviewer 3 Report

L23. Please reconsider "conditioning" as that term is used also for training. I suggest "competition season" and "post-season". Or, "in season" and off season" as stated in L59.

L100. The immediate analysis has nothing to do with the possibility of hemolysis. Please delete "so hemolysis is not considered a concern". In addition, please write in past tense.

L114. "pedaling was maintained continuously for 3 min". What was the rpm and load as post-exercise intensity may have affected the post-exercise lactate observations?

Ls 132, 133. Please clarify whether there was analysis of absolute value as it seems to indicate only changes were analysed. In addition, was there averaging of the lactate samples? Please clarify. Note this comment was not considered before.

L135. The one way anova was used to analyse absolute values. If you were analysing the changes from pre-season for in season and off season then a t-test would do to analyse the changes but that does not seem to be done.

L154-156. Please clarify as it seems the absolute values are analysed and not for example the change from pre-season values.

For headings 3.1 and 3.2 "Changes in" can be deleted as you are presenting the absolute values.

Author Response

We thank you for your kind comments on the resubmitted manuscript. We hope that all our corrections meet your criteria.

L23. Please reconsider "conditioning" as that term is used also for training. I suggest "competition season" and "post-season". Or, "in season" and off season" as stated in L59.

Thank you for your appropriate suggestion. We changed it to "in season" and "off season". (Line 22~23)

L100. The immediate analysis has nothing to do with the possibility of hemolysis. Please delete "so hemolysis is not considered a concern". In addition, please write in past tense.

Thank you. We have corrected this sentence as you suggested. (Line 101~102)

L114. "pedaling was maintained continuously for 3 min". What was the rpm and load as post-exercise intensity may have affected the post-exercise lactate observations?

Thanks, this is an important point. The load was set to turn off at the end of the Wingate test. We instructed the subject to continue pedaling to maintain blood circulation, but did not specify pedaling frequency. However, we observed that all subjects maintained a steady rhythm.

Ls 132, 133. Please clarify whether there was analysis of absolute value as it seems to indicate only changes were analysed. In addition, was there averaging of the lactate samples? Please clarify. Note this comment was not considered before.

Sorry for the lack of confirmation regarding the non-response to the comment. These measurements were analyzed in absolute values. Therefore, "Change in" has been removed. (Line 133)

L135. The one way anova was used to analyse absolute values. If you were analysing the changes from pre-season for in season and off season then a t-test would do to analyse the changes but that does not seem to be done

Since this experimental model had three periods of training cycles, we determined that the statistical procedure with one way anova was appropriate.

L154-156. Please clarify as it seems the absolute values are analysed and not for example the change from pre-season values

Corrections have been made as noted. “These were analyzed in absolute values.” (Line 157~158)

For headings 3.1 and 3.2 "Changes in" can be deleted as you are presenting the absolute values.

This description has been deleted. Thank you. (Line 153 ; 162)